# Intrathecal Injection of Autologous Mesenchymal Stem-Cell-Derived Extracellular Vesicles in Spinal Cord Injury: A Feasibility Study in Pigs

**DOI:** 10.3390/ijms24098240

**Published:** 2023-05-04

**Authors:** Ilya Shulman, Tatyana Ageeva, Alexander Kostennikov, Sergei Ogurcov, Leysan Tazetdinova, Ilyas Kabdesh, Alexander Rogozhin, Ilnur Ganiev, Albert Rizvanov, Yana Mukhamedshina

**Affiliations:** 1Neurosurgical Department No. 2, Republic Clinical Hospital, 420138 Kazan, Russia; 2Center for Clinical Research for Precision and Regenerative Medicine, Institute of Fundamental Medicine and Biology, Kazan Federal University, 420008 Kazan, Russia; 3Department of Morphology and General Pathology, Institute of Fundamental Medicine and Biology, Kazan Federal University, 420008 Kazan, Russia; 4Department of Neurology, Kazan State Medical Academy-Branch Campus of the Federal State Budgetary Educational Institution of Father Professional Education, Russian Medical Academy of Continuous Professional Education, 420012 Kazan, Russia; 5Scientific and Educational Center of Pharmacy, Kazan Federal University, 420008 Kazan, Russia; 6Department of Histology, Cytology and Embryology, Kazan State Medical University, 420012 Kazan, Russia

**Keywords:** spinal cord injury, mesenchymal stem cells, extracellular vesicles, pig

## Abstract

Spinal cord injury (SCI) remains one of the current medical and social problems, as it causes deep disability in patients. The use of mesenchymal stem cell (MSC)-derived extracellular vesicles (EVs) is one strategy for stimulating the post-traumatic recovery of the structure and function of the spinal cord. Here, we chose an optimal method for obtaining cytochalasin B-induced EVs, including steps with active vortex mixing for 60 s and subsequent filtration to remove nuclei and disorganized inclusions. The therapeutic potential of repeated intrathecal injection of autologous MSC-derived EVs in the subacute period of pig contused SCI was also evaluated for the first time. In this study, we observed the partial restoration of locomotor activity by stimulating the remyelination of axons and timely reperfusion of nervous tissue.

## 1. Introduction


The pathophysiology of SCI is complex and includes the death of neurons, the rupture of axons and secondary changes involving interrelated processes, such as blood–brain barrier dysfunction, local inflammation, and demyelination [1,2]. Currently, the clinically used methods of SCI therapy are not effective enough, and the search for such methods continues. Research over the past few decades has led to the creation of several new treatment strategies using extracellular vesicles (EVs). This is a promising approach to stimulate the regeneration of tissues, including the nervous system. EVs form a heterogeneous family of extracellular nanovesicles comprising exosomes, extravesicles, and apoptotic bodies [3]. Exosomes are the smallest type of EVs, with a diameter ranging from 40 to 150 nm, and are formed from the inward budding of endosomal membranes [4,5]. Apoptotic bodies are larger, with a diameter ranging from 50 to 5000 nm, and are formed by the fragmentation of cells undergoing programmed cell death [5,6]. EVs are larger than exosomes but smaller than apoptotic bodies, with a diameter ranging from 100 to 1000 nm, and are formed by the outward budding of the plasma membrane [7,8]. The function of exosomes and EVs is similar in that they both serve as vehicles for intercellular communication, as they are capable of transferring biologically active molecules, such as proteins, lipids, and nucleic acids, including miRNAs and mRNAs, from donor cells to recipient cells. The transferred molecules can affect various cellular processes, such as proliferation, differentiation, and apoptosis. They have been implicated in the regulation of the immune system, as well as in the pathogenesis of several diseases, including cancer, cardiovascular diseases, and neurological disorders. The main difference between exosomes and EVs is their biogenesis and cargo. Exosomes are formed from the inward budding of endosomal membranes and contain specific biomolecules, including proteins, lipids, and nucleic acids, that are selectively sorted and packaged into them [8,9,10]. EVs, on the other hand, are formed by the outward budding of the plasma membrane and contain a broader range of cellular components, including cytoplasmic proteins and lipids [11].

The use of EVs as therapeutic agents is an area of intense research. EVs can be isolated from various sources, including stem cells, blood, and urine, and can be used for drug delivery or as cell-free therapies. For example, EVs derived from mesenchymal stem cells (MSCs) have been shown to have regenerative properties and can be used to promote tissue repair in various diseases. Adipose tissue is the most commonly used source of mesenchymal stem cells (AT-MSCs) due to its accessibility and the convenience of obtaining it. AT-MSC-derived EVs promote intercellular communication and are involved in many physiological and pathological processes. The most significant function of AT-MSC-derived EVs is associated with their regenerative potential in the regulation of angiogenesis and anti-inflammatory responses within the inflammatory microenvironment in model of various neurological diseases in vitro [9,12]. More importantly, AT-MSC-derived EVs were able to significantly promote the recovery of neurological function, successfully reduce inflammation and promote neuroregeneration in rats after SCI [13,14,15,16]. Although these achievements have garnered significant scientific and clinical attention, there are certain barriers that limit their clinical translation. Two of the biggest challenges include scaling up EV production and the enhancement of the drug-loading capacity [17,18,19]. Several methods for the efficient production of EVs have been developed, including cytochalasin B-induced EVs, microfluidic fabrication, and serial extrusion through filters with decreasing pore sizes [20].

It is worth noting that, unlike EVs, which are designed to transport molecules synthesized by the cells themselves, “artificially produced” EVs can be made in different sizes and loaded with a variety of therapeutic molecules. In addition, methods for producing “artificial” EVs are more efficient when using the same number of cells, saving time for each production run. This group of methods includes a serial extrusion through filters with decreasing pore sizes [21], microfluidic fabrication [22], the “spin cups” method [23], the production of “ghost cells” and their further “squeezing” [24], and also cytochalasin B-induced EVs. Cytochalasin B-induced membrane vesicles contain functionally active surface proteins, the cytoplasmic component of parent cells, and maintain cellular signaling reactions [25]. A critical factor in the efficient production of EVs is the use of mechanical action, such as vortexing and filtering.

In our study, we utilized cytochalasin B-induced methods for obtaining EVs from AT-MSCs using different protocols for mechanical action, vortexing, and filtering. We then performed a qualitative assessment of the EVs. Additionally, we compared the long-term functional and structural outcomes of repeated intrathecal injection of autologous MSC-derived EVs in a pig spinal contusion model.

## 2. Results

### 2.1. Analysis of AT-MSC-Derived EVs by Electron Microscopy

We conducted an electron microscopic analysis of EVs derived from AT-MSCs using active vortexing for 30 s (1st), with (3rd) or without (2nd) post filtering. In the first protocol, we observed a destruction of the actin cytoskeleton and disintegration of the cytoplasm into many membrane-bound compartments after active mixing on the vortex for 30 s. However, the cell membrane and nuclei of AT-MSCs remained intact, as did the EVs. We also observed mitochondria and disorganized, cytoplasmic, non-membrane-bound inclusions. In the second protocol, without filtration and using active mixing on the vortex for 60 s, we observed a large scatter in the size and shape of AT-MSC-derived EVs. The sediment contained cell nuclei as well as disorganized cell contents without the membranes (Figure 1A). The cytoplasmic membranes of AT-MSCs were not preserved, and the EVs were divergent. The EVs without filtration were larger than those with subsequent filtration, and had contents of varying electron density. Some of the EVs contained multiple multilamellar bodies in the cytoplasm (Figure 1A’). In the third protocol, using active mixing on the vortex for 60 s with filtration, the sediment did not contain nuclei and disorganized inclusions. In this case, we found only the EVs (Figure 1B). The EVs from the third protocol were more uniform in size and smaller, in comparison to the second protocol, but they varied in form and content. We found circular mitochondria and individual multilamellar bodies in the composition of EVs (Figure 1B, Table 1). We also assessed the diameter of the resulting AT-MSC-derived EVs in the developed protocol (Figure 1C). Based on these results, subsequent studies and their use in pig SCI were carried out on EVs, obtained using the second protocol, by active mixing on the vortex for 60 s with filtration.

### 2.2. Cytokine Expression Profile of EVs

We found a significant increase in the levels of cytokines in the EVs when compared to the mesenchymal stem cell (MSC) supernatant, from which extracellular vesicles were derived, as a control: GM-CSF, IFN-g, IL-18, IL-2, IL-4 and IL-8 (Figure 1D; Appendix A). The maximum increase was found in the expression of IFN-g and IL-18, respectively.

### 2.3. Behavioural Outcomes

PTIBS. To measure the recovery of gross locomotor performance, hind limb function was assessed using the porcine thoracic injury behavior scale (PTIBS) designed by [26]. Scoring was performed on a weekly basis post-injury (Figure 2A). Prior to the injury, all animals achieved a baseline score of 10, indicative of normal hind limb function and locomotion. Following the SCI, locomotor function was most severely impaired at 1 wpi, with mean PTIBS scores of 1 for both the SCI and the SCI + EVs groups. We observed a significant rise of PTIBS scores in the SCI + EVs group from 9 wpi to the end of the experiment. By 12 wpi, the mean PTIBS scores were 2.2 and 5 for the SCI and the SCI + EVs groups, respectively. We also observed complete or partial dislocation of the hip joints in the animals of the SCI group but not in the pigs of the SCI + EVs group, which indicates a significant therapeutic effect of EVs.

Porcine neurological motor [PNM] score. Testing of hind limb clearance was only performed in the pigs from the SCI and the SCI + EVs groups at 1 to 12 wpi (Figure 2B). The test results of both groups showed a significant degradation in neurological motor score at 1 wpi, which was 0.6 and 0.8 scores for SCI + EVs and SCI, respectively. A significant difference between the groups was found at 6, 9 and 12 weeks, with the highest (2.5 folds) in the SCI + EVs group.

### 2.4. Electrophysiology

We recorded and characterized electrophysiological parameters, including M-wave, MEPs, and SSEPs, before and after the SCI at 6 and 12 weeks post injury (wpi). No significant differences in M-wave amplitude and latency were observed before and after injury. MEPs from the anterior tibial muscle were recorded in both legs of the intact animals. In the SCI + EVs group, most of the animals showed unilateral MEPs at 6 wpi, and two pigs showed MEPs only on one side at 12 wpi. It is worth noting that MEPs had a greater latency in the SCI + EVs group, compared to the initial values. In the SCI group, MEPs from the anterior tibial muscle were recorded only in one pig on one side at 6 and 12 wpi. Scalp and lumbar SEPs from the anterior tibial muscle were found in the intact animals. Scalp peaks were not recorded in both experimental groups at 6 and 12 wpi. The complex recorded from the lumbar level remained intact in experimental groups with the SCI. However, there was a significant decrease in the middle of N1 in animals without treatment in the SCI group compared to the intact control and the SCI + EVs group.

### 2.5. Routine Blood Examination EVs

We conducted routine blood examinations in the animals before and on the 3rd day post-injury (dpi), 1, 3, 6, and 8 wpi (Figure 3). In general, peripheral/venous blood values were within the normal range, previously identified in Guizhou minipigs [27]. Our study showed that the intrathecal injection of EVs resulted in a significant decrease in the number of leukocytes, erythrocytes, and hemoglobin at 3 wpi (Figure 3A,D,E). We also found a decrease in the number of lymphocytes (Figure 3B) in the EVs group by 1.8 times (*p* < 0.05), compared to the control group at 3 dpi. At the same time, the PLT value (Figure 3C), on the contrary, was increased 1.3 times in the SCI + EVs group compared to the SCI group. It is noted that the leukocyte and lymphocyte values were normalized in the SCI + EVs group by 8 wpi, while in the SCI group, these values were below the previously identified normal range. It is worth noting that the HGB value was significantly below the norm for all pigs, including before the experiments.

### 2.6. Quantification of the Spared Tissue and Abnormal Cavities

In both experimental groups with the SCI, a violation of the integrity of the nervous tissue in the area of injury and adjacent rostral and caudal areas at 12 wpi was found (Figure 4A). During the SCI, the gray matter was damaged, in which chromatolysis, apoptosis of neurons, and glial cells are visualized. It is worth noting that damage affects the gray matter to a greater extent than the white, which is clearly seen at a distance of 0.5 to 1 cm caudally from the epicenter of the SCI. No significant differences in morphometric analysis were found between the experimental groups. However, our results showed that the area of the spared tissue was increased by 27 percent, and the total area of abnormal cavities decreased by 29 percent in the SCI + EVs group in the caudal direction (from 0.5 to 1 cm), compared to the SCI group (Figure 4B,C).

### 2.7. Histological Evaluation of Myelin-Forming P0${}^{+}$-Cells and Vascularization

We showed the expression of myelin protein zero (P0) in the spinal cord of a pig after the injury. At the same time, two different structures with the P0 antigen were found on the sections: (1) myelin-associated structures, exhibiting a dense network of dotted and strand-like structures, apparently formed by oligodendrocytes, (2) as well as myelin around individual fibers, probably formed by migrating Schwann cells (Figure 5A). A population of myelin-forming P0${}^{+}$-cells was identified in the dorsal root entry zone (DREZ) at the distance of 0.5 cm caudal to the lesion epicenter. We showed that injections of EVs contribute to an increased (*p* < 0.05) number of P0+-cells in the DREZ, compared to the SCI group without the therapy (Figure 5C). An immunofluorescent assay of vascularization with antibodies to CD31 demonstrated that nervous tissue was, on average, better perfused after the EVs injection, compared to the SCI group (Figure 5B). Injections of EVs contributed to the activation of angiogenesis and the stabilization of newly formed vessels after the porcine SCI. In the aforementioned distance from injury, the number of CD31${}^{+}$-cells was 2.5-fold higher (*p* < 0.05) in the SCI + EVs group than in the SCI group at 12 wpi (Figure 5D).

**Figure 4 ijms-24-08240-f004:**
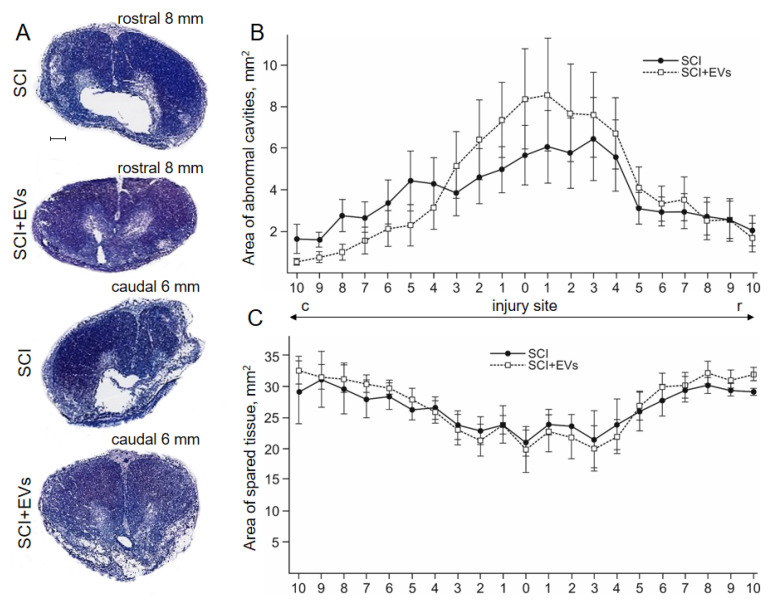
Tissue analysis in the experimental groups. (**A**) Cross sections of the injured spinal cord at 12 weeks post-injury of pigs in the experimental groups with azur–eosin staining. Scale bar 0.2 mm. An area of a total area of abnormal cavities (**B**) and the spared tissue (**C**) within 10 mm caudally and rostrally from the injury epicenter at 12 weeks post-injury of pigs. No significant differences in the above indicators of morphometric analysis were found between the experimental groups, one-way ANOVA followed by a Tukey’s post hoc test.

## 3. Discussion

SCI remains one of the current medical and social problems, as it causes deep disability in patients. Despite significant efforts to improve the functional outcome of SCI patients, secondary complications, characterized by an inflammatory response and apoptosis in the central nervous system, remain the main cause of neurological dysfunction [28]. The use of EVs is one strategy for stimulating post-traumatic recovery of the structure and function of the spinal cord. There are numerous methods for obtaining EVs, but it is important to note that in most cases it is impossible to achieve complete separation of the desired vesicles [11]. We proposed a cytochalasin B-induced method for obtaining EVs from pig AT-MSCs and conducted a comparative evaluation, using electron microscopy with modified protocols. In our opinion, the most optimal method for obtaining EVs is the protocol that involves active vortex mixing of EVs for 60 s and subsequent filtration to remove nuclei and disorganized inclusions on comparative evaluation, using electron microscopy with modified protocols. This protocol was found to have good reproducibility and produced more uniform and smaller EVs without nuclei or disorganized inclusions, with circular mitochondria and individual multilamellar bodies in their composition. In contrast, the protocol with the active vortexing of cells for only 30 s did not achieve complete separation of EVs and carried risks due to the presence of an intact cell nuclei. On the other hand, the protocol that was similar to the optimal method with active vortexing of cells for 60 s but included post-filtering produced EVs with significant variation in size and shape.

We found that EVs, obtained from AT-MSCs, had a stable level of cytokine expression, capable of regulating immune and inflammatory responses. It is important to note that the comparison of cytokine levels in EVs versus MSC supernatant provided evidence of the selective packaging of cytokines into EVs. The observed increase in the expression of IFN-g and IL-18 in EVs may have important implications for their therapeutic potential, as both cytokines have been shown to play critical roles in modulating immune responses, and have been investigated as potential therapies for various diseases. IL-18, as the main immunoregulatory cytokine, plays an important role as a factor in anti-infectious and anti-tumor protection of the body [29]. Additionally, the significant increase in the levels of other cytokines, including GM-CSF, IL-2, IL-4, and IL-8, suggests that EVs from AT-MSCs may have a broad range of immunomodulatory effects, which could be beneficial in the context of a SCI. GM-CSF has been shown to promote neuronal survival and differentiation in vitro and to improve functional recovery after spinal cord injury in rats [30,31]. IL-2 has been reported to have both neuroprotective and neurotoxic effects, depending on the context. In some studies, IL-2 was shown to promote neuronal survival and function [32,33], while in others it was reported to induce apoptosis and impair synaptic function [34]. IL-4 has been shown to have neuroprotective effects in several models of neurodegeneration, including Alzheimer’s disease and Parkinson’s disease [35,36]. IL-4 has also been reported to promote neuronal survival and differentiation in vitro [37]. IL-8 was shown to promote neuronal survival and function [38]. Such an effect of EVs from AT-MSCs can have a positive impact on populations of motor neurons in spinal cord injury, improving motor function.

Here, we demonstrated that repeated intrathecal administration of EVs, derived from AT-MSCs, was significantly effective in improving long-term functional outcomes. Based on PTIBS and PNMS scores, we showed that intrathecal infusion of EVs in the pigs stimulated their motor activity recovery up to 12 weeks post-SCI. However, electrophysiological analysis showed only slight improvement in motor-evoked potentials in some pigs with the EVs therapy. Previous studies by other researchers have already shown significant improvement in motor function in rats after the injection of MSC-EVs [26,39]. However, there are currently no studies with the injection of MSC-EVs in a pig SCI. Preserving or restoring the organization of neural tissue structure is of a paramount importance for its functionality. We demonstrated the long-term effect of intrathecal administration of EVs derived from AT-MSCs on the condition of tissue in the injury area, including the area of spared tissue and the total area of abnormal cavities. Despite the absence of significant differences between the experimental groups, based on these criteria, there are positive dynamics in the transplantation of EVs, compared to the control (SCI) group. Previously, it was shown that intravenous or intraspinal injection of MSC-EVs to rats after a SCI promotes improvement in the condition of neural tissue in a delayed period with better performance in the last group, which is due to a decrease in scarring and inflammatory reactions from microglial cells, as well as the greater preservation of neurons and their axons [40].

We demonstrated, for the first time, the effect of the intrathecal injection of EVs, derived from AT-MSCs, on the state of myelin-forming cells, expressing P0 protein in the pig spinal cord after the SCI. Numerous studies have emphasized the therapeutic effects of EVs, in terms of stimulating the proliferation and/or differentiation of oligodendrocytes, as well as remyelination and axon regeneration. For instance, in a study on prenatal brain injury, which affects both white and gray matter and has serious consequences for nervous system development, MSC-EV injection contributed to the preservation of myelination [41]. In another study, the above-mentioned therapy on a rat model of premature traumatic brain injury effectively improved inflammation-induced hypomyelination and long-term microstructural abnormalities in the white matter [42]. The destruction of blood vessels in the injured spinal cord is a serious consequence of SCI, given the critical importance of the blood supply to the spinal cord [43]. MSC-derived EVs promote HUVEC proliferation and migration, increase tubule formation capacity, and upregulate angiogenesis-related genes, such as VEGF and HIF-1α [44,45,46]. We showed the pro-angiogenic effect of AT-MSC-derived EVs. Using CD31 immunostaining, we demonstrated that MSC-EV treatment significantly increased the total number of blood vessels in the SCI lesion 12 weeks after injury. An increase in the number of blood vessels is necessary to ensure an adequate supply of oxygen and nutrients to damaged tissues, which can promote the survival and regeneration of neurons and glial cells [47], as well as contributing to the clearance of cellular debris and toxic molecules [48]. However, for future implementation in clinical practice, it is necessary to reveal in more detail the molecular mechanisms of angiogenesis, myelination, immunomodulation, and other processes in the treatment of AT-MSC-derived EVs.

## 4. Materials
and Methods

### 4.1. Cultivation of Mesenchymal Stem Cells and EVs Isolation

Pig adipose tissue-derived MSCs were obtained aseptically from the subcutaneous fat of 3–4-month-old female pot-bellied pigs under anesthesia (IV, 2–6 mg/kg, Fresenius Kabi, Bad Homburg, Germany) via endotracheal intubation and maintained with isoflurane (1.3%, Laboratorios Karizoo, Barcelona, Spain). The obtained subcutaneous fat was collected in sterile 50 mL tubes and delivered to the culture laboratory for subsequent cultivation. The adipose tissue was thoroughly homogenized with sterile scissors, and 0.9% NaCl solution was added and then centrifuged at 500× *g* for 5 min, after which the supernatant was removed. A freshly prepared sterile solution 0.2% crab hepatopancreas collagenase (Biolot, Saint Petersburg, Russia) was added to the adipose tissue homogenate and incubated at 37 °C for 1 h, with constant shaking at 180 rpm (on a rocking platform). Then, the homogenate was centrifuged at 500× *g* for 5 min, and the supernatant removed. The resulting cells of stromal vascular fraction of adipose tissue were seeded into culture flasks and cultivated in medium containing DMEM, 20% FBS, 2 mM l-glutamine, 100 µM l-ascorbic acid 2-phosphate, 100 U/mL penicillin, and 100 mg/mL streptomycin (all obtained from PanEco, Moscow, Russia) at 37 °C and 5% CO_2_. The non-adherent cells after 24–48 h were removed, and the culture medium was replaced every 3 days. The third passage cells were used to obtain EVs. The obtained MSCs were cultured to a monolayer density of 90–95%. The culture medium was then removed, and cells were washed in DPBS and transferred to suspension by treatment with 0.25% trypsin solution with subsequent inactivation by adding the DMEM medium containing 10% FBS. Cells were then centrifuged in falcons for 5 min at 1400 rpm and washed in 0.9% NaCl to remove serum residues. After, cells were incubated in serum-free DMEM medium containing 10 μg/mL cytochalasin B (Sigma-Aldrich, Burlington, MA, USA) for 30 min (37 °C, 5% CO_2_) (Figure 6). At the end of an incubation, the cell suspension was actively vortexed for 30 Section (1st method) and 60 Section (2nd method) and pelleted (50 g for 10 min). The obtained supernatant was subjected to two subsequent centrifugation steps (100× *g* for 10 min and 8000× *g* for 15 min) and further filtration using a sterile filter (0.45 μm, GVS Filter Technology, Morecambe, UK) (3d variant with pre-mixing on a vortex for 60 s). The resulting pellet contained EVs was resuspended in 0.9% NaCl.

### 4.2. Electron Microscopic Analysis of EVs from AT-MSCs

To assess the ultrastructure of the obtained EVs, we fixed it in 2.5% glutaraldehyde. After 12 h from the start of fixation, the samples were placed in a 1% OsO_4_ solution in a phosphate buffer with added sucrose, dehydrated and embedded in Epon-812 (Fluka, Charlotte, NC, USA). Ultrathin sections of 0.1 μm thickness were prepared on a Leica UC7 ultramicrotome (Leica, Wetzlar, Germany) and mounted on copper grids (Sigma). Sections were counterstained with uranyl acetate and lead citrate and then examined using a Hitachi 7700 transmission electron microscope (Hitachi, Tokyo, Japan).

### 4.3. Cytokine Assay

During the study, a multiplex analysis was carried out using the xMAP Luminex technology of the the EVs from AT-MSCs (10 μg in 25 μL of 0.9% NaCL). MILLIPLEX MAP kit Porcine Cytokine/Chemokine (magnetic) kit PCYTMG-23K-13PX (Merck, Darmstadt, Germany) was used. There was a multiplex analysis of 13 cytokines/chemokines/interleukins (GM-CSF, IFN-g, IL-1a, IL-1b, IL-1Ra, IL-2, IL-4, IL-6, IL-8, IL-10, IL-12, IL-18, TNF-a) in accordance with the requirements of manufacturers.

### 4.4. Animals

Experiments were carried out on pot-bellied pigs, mature females weighing 8–10 kg and aged 3–4 months. The animal study was reviewed and approved by the local ethics committee of Kazan Federal University (No. 2, 5 May 2015). Efforts were made to minimize the number of animals used and their suffering. The surgical procedures, post-operative procedures, daily care and termination procedures were supervised by veterinarians.

### 4.5. Spinal Cord Injury and Intrathecal Injection

All invasive procedures with the animals were performed under intubation anesthesia, appropriate pre-operation preparation, and adequate analgesia/pain control. Premedication was carried out using a combination of medetomidine (0.06–0.07 mg/kg, Apicenna, Moscow, Russia) and zolazepam (2–3 mg/kg, Zoletil 100, Virbac Sante Animale, Carros, France). After a propofol induction (IV, 2–6 mg/kg), endotracheal intubation with cuffed endotracheal tubes (size 4.0–4.5) was performed using isoflurane (2–3%) and zolazepam (3–6 mg/kg/h) throughout the intervention. Weight-drop injury, which inflicts a severe contusion injury to the spinal cord, was induced using an impact rod (weight of 50 g, diameter of 6 mm) that was dropped from a height of 40 cm, followed by compression with 100 g weight for 5 min, after laminectomy at the Th11 level. A urinary catheter (10 Fr, Jorgensen Laboratories Inc., Loveland, CO, USA) was inserted 3–5 days after surgery. Cefazolin (25 mg/kg, Sintez, Kurgan, Russia) and ketoprofen (1 mg/kg, AVZ, Moscow, Russia) were given as intramuscular injections. The pigs were housed separately within the first 48 h, then in pairs. After 1 wpi (week post-injury) the paraplegic pigs were randomly divided into two groups. After 1 and 3 wpi, the animals (treated group, n = 6) received an intrathecal injection (L4–L5) 300 μg of MSC-derived EVs in 300 μL of 0.9% NaCl at a rate of 30 μL/min, using a microinjection pump (Syringe One, New Era Instruments, NY, USA) for each animal. A two-stage EVs injection option was chosen to augment the beneficial effects. The pigs (n = 3, control group) were subjected to the same protocols, using the same volumes and rates of 0.9% NaCl injection. The pigs were under round-the-clock veterinary supervision of all vital signs, including body temperature, pulse and respiration rates and blood pressure. During the EV injections and also in the post-injection periods, we did not observe any adverse events from the above indicators, such as vomiting, diarrhea, seizures, and other symptoms.

### 4.6. Porcine Thoracic Injury Behavioral Scale

To evaluate the effectiveness of a locomotor function recovery, the Porcine Thoracic Injury Behavioral Scale (PTIBS) was used [49]. The PTIBS is a 10-point scale that describes various stages of hind limb function. Locomotor recovery in the study groups was video recorded as previously described [50]. The tests were conducted beginning from the third day after surgery once a week till the end of the experiment, by 12 wpi.

### 4.7. Porcine Neurological Motor Score

Locomotor function was also assessed using a 14-point porcine neurological motor (PNM) scoring system [51]. Animals were allowed to walk freely in an open space, and hind limb movement was observed for 5 min and assigned a score between 0 (no observable movement in either hind limb and tail) and 14 (capable of standing up spontaneously on hind limbs with sustained locomotion; consistent plantar-hoof stepping; consistent forelimb–hind limb coordination; able to pass hind limbs clearance test; and tail movement present).

### 4.8. Electrophysiological Studies

The animal’s neuromotor function was assessed by stimulating electromyography as previously described before the injury [50,52], and at 6 and 22 weeks after the SCI. M-waves were recorded from the tibialis anterior muscle in response to a stimulation of the sciatic nerve. Monopolar needle electrodes were used for recording and reference. Transcranial electrical stimulation (TES) was used for the evaluation of the pyramidal tracts. Motor-evoked potentials (MEPs) were registered from the tibialis anterior muscle by needle electrodes inserted under the scalp up to contact with the skull bone. Somatosensory evoked potentials (SEPs) were registered for the evaluation of spinal cord posterior columns, using monopolar needle electrodes, which were subcutaneously inserted.

### 4.9. Complete Blood Count Analysis

Venous blood samples were obtained from the ear veins of pigs before the SCI, at 3 dpi and 1, 3, 6, 8 wpi. The complete blood count was analyzed in Hematology Abacus Junior 5 Vet (Diatron Messtechnik GMBH, Austria), with EDTA-K2 as the anticoagulant. This analysis included the following variables: hemoglobin (HGB, g/L), red blood cell (RBC, 10${}^{12}$/L), platelet (PLT, 10${}^{9}$/L), white blood cell (WBC, 10${}^{9}$/L), and lymphocyte (LYM, 10${}^{9}$/L).

### 4.10. Histological Procedure

At 12 wpi, the animals were anesthetized and perfused with a 4% paraformaldehyde solution (4 °C). A fragment of the spinal cord (3 cm) was taken from the spinal column and fixed in a 4% paraformaldehyde solution for 2 days. Then, the sample was transferred into 30% sucrose. Cryostat cross sections of the spinal cord over 1 cm from the injury epicenter rostrally and caudally were stained with azur–eosin (MiniMed, Bryansk, Russia). The stained sections were embedded into vitrogel and studied under the APERIO CS2 scanner (Leica, Deer Park, IL, USA). During the study, the area of spared nervous tissue and the total area of abnormal cavities were estimated using the Aperio ImageScope 12.4 software (Leica) for morphometric analysis.

### 4.11. Immunofluorescence Analysis

Immunofluorescence reactions were conducted in a standard way [52]. The sections were incubated with primary and secondary antibodies to identify the antigen: P0 (Santa Cruz, 1:50), CD31 (Abcam, Cambridge, UK, 1:100), donkey anti-mouse Alexa Fluor 555 (Abcam, 1:200), and donkey anti-rabbit Alexa Fluor 647 (Abcam, 1:200). Tissue sections were observed using a LSM 700 confocal microscope (Carl Zeiss, Oberkochen, Germany). The number of immunopositive cells was blindly quantified in the dorsal root entry zone (DREZ) in merged images from 10 adjacent optical slices (512 × 512 pixel resolution, observed area 0.05 mm^2^; acquisition distance, 0.5 μm). Only the cells with clearly outlined nuclei by DAPI (10 µg/mL in PBS, Sigma) were considered. Negative controls were obtained using the same protocol but without the addition of primary or secondary antibodies. Digital images of the sections of the spinal cord were analyzed using the software ImageJ 1.47j (http://imagej.nih.gov/ij, accessed on 15 March 2021).

### 4.12. Data Analysis

Data are expressed as mean ± standard error of the mean (SEM). To determine statistical significance, we used a Student’s *t*-test distribution or a one-way analysis of variance (ANOVA) with Tukey’s test. A value of 0.05 was considered statistically significant. All analyses were performed in a “blinded” manner with respect to the treatment group. Data were analyzed using Origin 7.0 SR0 (OriginLab, Northampton, MA, USA) software.

## 5. Conclusions

AT-MSC-derived EVs could potentially become a convenient and readily available source for use in clinical practice. However, there are still unresolved issues, related to scaling production and obtaining stable contents of EVs. In this study, the therapeutic potential of intrathecal injection of AT-MSC-derived EVs in the subacute period of pigs, contused SCI, was evaluated for the first time. The results showed that this approach could help improve locomotor activity by stimulating the remyelination of axons and timely reperfusion of nervous tissue. However, significant structural improvement or a full restoration of function in the injured pig spinal cord through EVs therapy was not observed. Nonetheless, given the severe degree of the SCI model, this approach of regenerative medicine cannot be ignored.

## Figures and Tables

**Figure 1 ijms-24-08240-f001:**
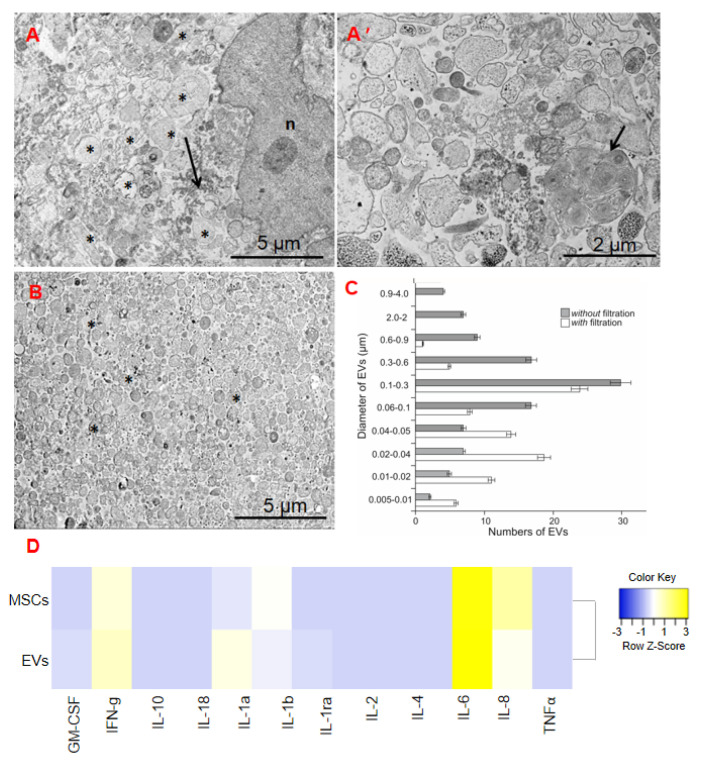
Transmission electron microscopy and cytokine profile analysis of AT-MSC-derived EVs. (**A**) EVs (asterisks) were obtained without the filtration, using active mixing on the vortex for 60 s. There is a very large variation in size and shape, sediment contains the nucleus (n) and disorganized cell contents, that are not limited by the membranes (arrow). However, the cytoplasmic membranes of MSCs are not preserved and the vesicles diverge. (**A’**) EVs, obtained without the filtration, have a variable contents with different electron density, some enclosing multiple multilamellar bodies (arrow). (**B**) Sediment, obtained using active mixing on the vortex for 60 s with the filtration, does not contain nuclei and disorganized inclusions. Only the EVs (asterisks) are present in the sediment, which are more uniform in size (smaller than without filtration), but vary in shape and content. (**C**) Quantitative distribution of EVs diameter in case of active mixing on the vortex for 60 s with and without filtration. (**D**) Graphical representation showing cytokine concentrations (color keys), generated with the multiplex analysis of EVs and mesenchymal stem cells (MSCs) supernatant as a control.

**Figure 2 ijms-24-08240-f002:**
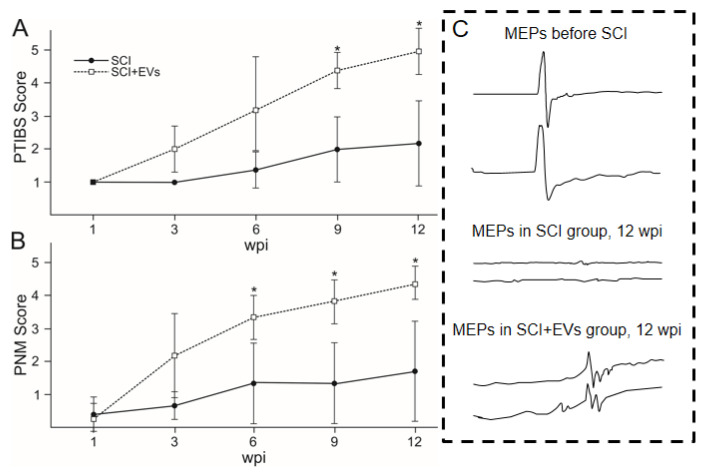
Results of functional tests on the (**A**) porcine thoracic injury behavior scale (PTIBS) and (**B**) porcine neurological motor (PNM) scores, which were measured before injury and weekly for 12 weeks (84 dpi) post injury for SCI and SCI + EVs groups. Data points represent mean ± SEM for n = 5 animals per group. * *p* < 0.05. (**C**) The electrophysiology results show motor-evoked potentials (MEPs) before the SCI and at 12 wpi in SCI and SCI + EVs groups.

**Figure 3 ijms-24-08240-f003:**
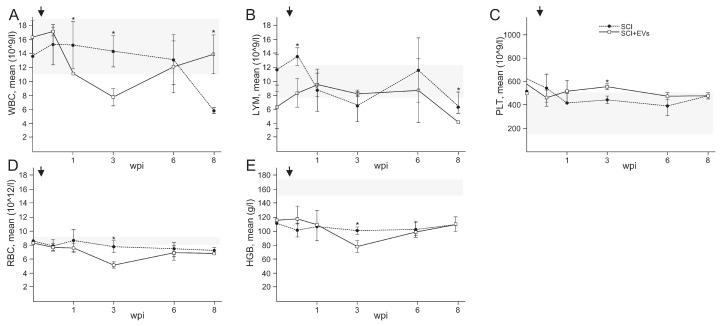
Dynamics of routine blood examination of pigs in SCI and SCI + EVs groups: (**A**) WBC (white blood cells), (**B**) LYM (lymphocyte), (**C**) PLT (platelet), (**D**) RBC (red blood cells), (**E**) HGB (hemoglobin). Gray line indicates peripheral/venous blood values for previously identified in Guizhou minipigs. Arrows indicate 3 dpi. Data points represent mean ± SEM for n = 5 animals per group. * *p* < 0.05.

**Figure 5 ijms-24-08240-f005:**
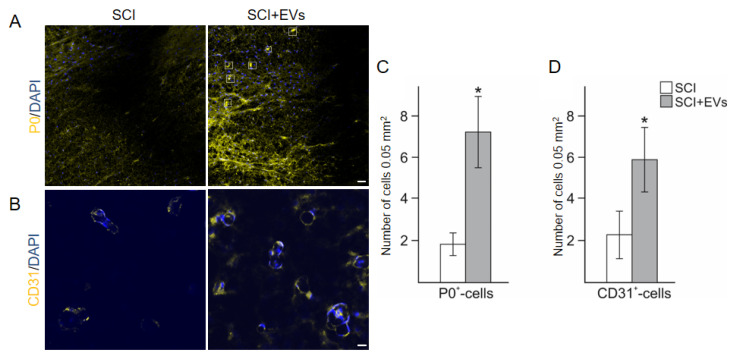
Visualization of P0 ((**A**), yellow) and CD31 ((**B**), yellow) expression 5 mm caudally from the injury epicenter within the DREZ in the investigate groups. Nuclei are DAPI-stained (blue). Scale bar = 20 (**A**) and 10 (**B**) µm. Selected squares demonstrated myelin, probably formed by migrating Schwann cells. Number of P0${}^{+}$-Schwann cells (**C**) and CD31${}^{+}$-cells (**D**) in SCI (white column) and SCI + EVs (grey column) groups at 12 wpi. Data points represent mean ± SEM for n = 5 animals per group. * *p* < 0.05.

**Figure 6 ijms-24-08240-f006:**
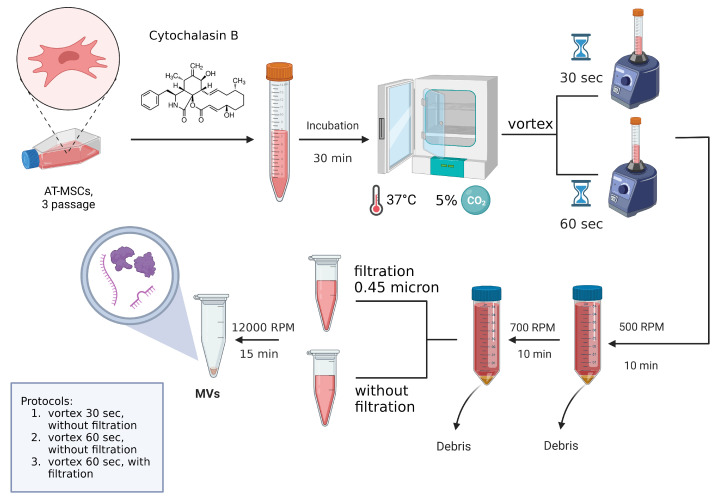
Graphical summary of used protocols of EVs isolation.

**Table 1 ijms-24-08240-t001:** Characterization of obtained AT-MSC-derived EVs based on electron microscopy analysis.

Variants of Protocol	Nuclei of MSCs	Disorganized Non-Membrane-Bound Inclusion	Mitochondria	Multilamellar Bodies in the Cytoplasm	Shape	Size, μm ^1^
1st protocol (active vortexing of cells for 30 s)	Yes	Yes	Yes	multiple	N/A	N/A
2nd protocol (active vortexing of cells for 60 s without post-filtering)	Yes	Yes	Yes	multiple	varies	0.06–0.6
3rd protocol active vortexing of cells for 60 s with post-filtering)	No	No	Yes	individual	mostly round	0.02–0.3

^1^ An EVs size range is specified that is greater than 55% of the EVs received. N/A—in the case of active vortexing of cells for 30 s, the EVs remained in the cells and did not disperse.

## Data Availability

The data presented in this study are available on request from the corresponding author. The data are not publicly available due to the evolving nature of the project.

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
