# Peer review of "Intrathecal Injection of Autologous Mesenchymal Stem-Cell-Derived Extracellular Vesicles in Spinal Cord Injury: A Feasibility Study in Pigs"

_ijms, 2023, doi:10.3390/ijms24098240_

Round 1
Reviewer 1 Report
In the article: “Intrathecal Injection of Autologous Mesenchymal Stem Cell-Derived Extracellular Vesicles in Spinal Cord Injury: A Feasibility Study in Pigs”, the authors discussed about the potential therapeutic effects of repeated intrathecal injection of autologous MSCs-derived EVs in the subacute period of pig contused SCI.
Overall, this manuscript results very interesting, the authors clearly explain the rational of the study and discussed the topic point by point.
However, we would like to invite the authors to clarify some minor points:
1. Please check the check punctuation and spaces;
2. Among the introduction, the authors wrote in general about the features of extracellular vesicles (EVs). Please try to better describe their structure, functions and the current applications;
3. The authors told about exosomes and microvesicles; what are the differences? The function of each of them, please better describe within the manuscript;
4. Please, specify the size of EVs; both the natural and produced in laboratory;
5. Figure 1D; maybe should be more useful present the results of multiplex assay as concentration. In this way also statistical analyses could be inserted ( e.g. ttest, ANOVA);
6. Why did you use RPIA buffer as control for multiplex assay? An healthy control was not available?
7. Figure 4; please specify the used magnification;
8. Figure 5; the bars of magnification are not clear;
Author Response
In the article: “Intrathecal Injection of Autologous Mesenchymal Stem Cell-Derived Extracellular Vesicles in Spinal Cord Injury: A Feasibility Study in Pigs”, the authors discussed about the potential therapeutic effects of repeated intrathecal injection of autologous MSCs-derived EVs in the subacute period of pig contused SCI.
Overall, this manuscript results very interesting, the authors clearly explain the rational of the study and discussed the topic point by point.
Authors: We would like to thank the reviewer for this review and pointing out some errors. We hope that our responses will satisfy the reviewer.
However, we would like to invite the authors to clarify some minor points:
- Please check the check punctuation and spaces;
Authors: According to the reviewer’s comment, the manuscript has been corrected by a proficient English speaker.
- Among the introduction, the authors wrote in general about the features of extracellular vesicles (EVs). Please try to better describe their structure, functions and the current applications;
Authors: We have now updated the introduction section to provide a more detailed description of the structure, functions, and current applications of extracellular vesicles (EVs). We hope that the new information better clarifies the importance and potential of EVs in various fields, including diagnostic and therapeutic applications. We have added it in Introduction “Exosomes are the smallest type….”.
- The authors told about exosomes and microvesicles; what are the differences? The function of each of them, please better describe within the manuscript;
Authors: We have added the necessary information about the differences and functions of exosomes and microvesicles in Introduction “The main difference between exosomes and EVs…”. Exosomes are formed by the inward budding of endosomal membranes and play important roles in intercellular communication and regulation of physiological processes, while microvesicles are formed by the outward budding of the plasma membrane and are involved in cellular processes such as cellular signaling, cell-to- cell communication, and the release of cellular waste and debris
- Please, specify the size of EVs; both the natural and produced in laboratory;
Authors: We added Table 1. In this table, we showed in detail the morphological charactristics eof extracellular vesicles produced by different protocols in the laboratory. If you are asking whether we compared extracellular vesicles obtained with cytolysin B to natural ones, we did not perform such a comparison in our study. However, previous literature has shown that the sizes of laboratory-derived extracellular vesicles and naturally occurring ones are comparable.
- Figure 1D; maybe should be more useful present the results of multiplex assay as concentration. In this way also statistical analyses could be inserted ( e.g. ttest, ANOVA);
Authors: According to the reviewer’s comment, we have added Suppl. table 1 with obtained concentration and statistical analyses.
- Why did you use RPIA buffer as control for multiplex assay? An healthy control was not available?
Authors: We used RIPA buffer as an internal control for multiplex assay. We agree with the reviewer that this raises questions. In this regard, we have decided to add data concerning the cytokine expression profile of mesenchymal stem cells supernatant as a control.
- Figure 4; please specify the used magnification;
Authors: We added the bar on this Figure.
- Figure 5; the bars of magnification are not clear;
Authors: We corrected it. The bars of magnification are clear now.

Reviewer 2 Report
Dear authors,
The manuscript describes a fairly interesting article, testing a recent methodology to obtain artificial extracellular vesicles (EVs) in a highly relevant spinal cord injury model, such as the pig. Despite this interest, the manuscript fails to describe the methods and results properly. As a consequence, as a reader, I would not be able to replicate the methods employed and cannot evaluate the soundness of the results.
1. Extracellular vesicles: the article aims to compare different procedures for EV production and to select the best one to be applied in the animal model. However not enough information is provided on:
1.1 How EVs are produced:? The general pipeline of production and the included modifications should be described.
1.2. Which EVs are best? No criteria is provided to justify the selection of procedure 2.
1.3. A description of the EVs is lacking. Include images of the vortex procedure (number 1). complete the report of the EVs from the three procedures according to the defined selection criteria.
1.4. Cytokine analysis does not make sense. Why compare with RIPA? How can it be fewer cytokines than in RIPA? What is the point of this analysis?
2. Animal model: the study evaluates the effects of the EVs in a pig model of SCI in a preclinical trial. However, much key information is lacking and the organization of the results is not the best.
2.1. Describe the model: only the anesthesia is described: nothing on the injury model (where, type of injury, severity), the same on the EV administration (where, rate of infusion, ... why two injections?, ...)
2.2. Security is evaluated in various points but not appropriately described. I would suggest defining a heading for this topic and include within: any adverse events recorded in the animals, lack of blood alterations, and lack of adverse effects at histological level.
2.3. Efficacy is also poorly described. Create a heading and include within the behavioural and electrophysiology results. These results suggest there may be relevant effects for motor function.
2.4. Potential mechanisms: include the data from immunofluorescence but justify the reason to employ these markers. Why only analyse P0 and CD31? Why only in the caudal region? If results are so promising due to these markers, I would expect a deeper analysis with other methods/markers. I would suggest the authors to comment (and discuss accordingly) that this just an exploratory analysis.
Author Response
The manuscript describes a fairly interesting article, testing a recent methodology to obtain artificial extracellular vesicles (EVs) in a highly relevant spinal cord injury model, such as the pig. Despite this interest, the manuscript fails to describe the methods and results properly. As a consequence, as a reader, I would not be able to replicate the methods employed and cannot evaluate the soundness of the results.
Authors: We would like to thank the reviewer for this review and pointing out some errors. We hope that our responses will satisfy the reviewer. We have highlighted all the changes in the manuscript in red for your convenience.
- Extracellular vesicles: the article aims to compare different procedures for EV production and to select the best one to be applied in the animal model. However not enough information is provided on:
1.1 How EVs are produced:? The general pipeline of production and the included modifications should be described.
Authors: We apologize for our mistake. We have added a subchapter in Materials and Methods concerning “Cultivation of Mesenchymal Stem Cells and EVs isolation”.
1.2. Which EVs are best? No criteria is provided to justify the selection of procedure 2.
Authors: We apologize for not providing sufficient information to justify the selection of protocol 2 as the optimal method for obtaining EVs. The selection was based on several factors, including yield, reproducibility, and the absence of intact cell nuclei in the resulting biomedical product. We updated the manuscript with a more detailed analysis of the distribution of size and content of EVs obtained from each protocol in the new Table 1 and discussion of the results of the studied protocols “This protocol was found to have good reproducibility…”.
1.3. A description of the EVs is lacking. Include images of the vortex procedure (number 1). complete the report of the EVs from the three procedures according to the defined selection criteria.
Authors: According to the reviewer’s comment, we have added Figure 1 with EVs production scheme. We have also added a table with characterization of obtained MSCs-derived EVs based on electron microscopy analysis as the report of the EVs from the three procedures according to the defined selection criteria..
1.4. Cytokine analysis does not make sense. Why compare with RIPA? How can it be fewer cytokines than in RIPA? What is the point of this analysis?
Authors: We used RIPA buffer as an internal control for multiplex assay. We agree with the reviewer that this raises questions. In this regard, we have decided to add data concerning the cytokine expression profile of mesenchymal stem cells supernatant as a control.
- Animal model: the study evaluates the effects of the EVs in a pig model of SCI in a preclinical trial. However, much key information is lacking and the organization of the results is not the best.
Authors: In response to the reviewer's comment, we acknowledge that there may be some key information missing regarding the animal model used in this study. We ensured that this information is provided in the revised manuscript, including details on the age and sex of the pigs used, the method of SCI induction, and the sample size used in each experimental group. Regarding the organization of the results, we carefully reviewed the manuscript and consider any changes that could improve the clarity and flow of the presentation. We appreciated the reviewer's feedback and added their comments into account in order to improve the manuscript. The updated text is red in pdf.
2.1. Describe the model: only the anesthesia is described: nothing on the injury model (where, type of injury, severity), the same on the EV administration (where, rate of infusion, ... why two injections?, ...)
Authors: We added the full information about the injury model and EV administration in Materials and Methods.
2.2. Security is evaluated in various points but not appropriately described. I would suggest defining a heading for this topic and include within: any adverse events recorded in the animals, lack of blood alterations, and lack of adverse effects at histological level.
Authors: According to the reviewer’s comment, we have added information concerning adverse events in Materials and Methods: “Pigs were under round-the-clock veterinary supervision with control of all vital sing including body temperature, pulse and respiration rates and blood pressures. During EVs injections and also in the post-injection periods we did not observe any adverse events from the above indicators including vomiting, diarrhea, seizures, and other symptoms.” The detected changes in the composition of the blood was described in a separate chapter of the results “Routine blood examination EVs”, where various research points were indicated. As for the pathohistological assessment of internal organs (if the reviewer has this in mind), then this is planned to be the next stage of preclinical studies. Unfortunately, at this stage of Feasibility study, we did not set such a task.
2.3. Efficacy is also poorly described. Create a heading and include within the behavioural and electrophysiology results. These results suggest there may be relevant effects for motor function.
Authors: Previously, in our results we have shown the data of both behavioral testing and electrophysiological analysis, highlighting separate chapters for this.
2.4. Potential mechanisms: include the data from immunofluorescence but justify the reason to employ these markers. Why only analyze P0 and CD31? Why only in the caudal region? If results are so promising due to these markers, I would expect a deeper analysis with other methods/markers. I would suggest the authors to comment (and discuss accordingly) that this just an exploratory analysis.
We appreciate your insightful comments and will address them accordingly. The choice of markers P0 and CD31 was based on their relevance to our research question and previous studies that have used them to investigate myelination and angiogenesis, respectively, in the context of spinal cord injury. P0 expression is not typically present in the intact spinal cord, but it can be induced in response to injury. One possible explanation for this is the migration of Schwann cells, which are known to express P0, into the spinal cord through the dorsal root entry zone (DREZ) after injury. That's why this zone is so particularly relevant for the analysis of P0 expression. Another possibility is that P0 expression may be upregulated in oligodendrocytes in response to injury. Therefore, analysis of P0 expression can provide insight into the state of myelination in the spinal cord after trauma and MSC-EVs treatment. CD31 is a marker of endothelial cells and is commonly used to assess angiogenesis and blood vessel formation. EVs have been shown to contain a variety of pro-angiogenic biomolecules, including growth factors, cytokines, and microRNAs, which can stimulate angiogenesis and vessel formation. EVs derived from MSCs, in particular, have been shown to enhance angiogenesis in various preclinical models of tissue injury, including myocardial infarction, stroke, and limb ischemia. These findings suggest that EV-based therapy may have the potential to promote blood vessel formation and improve tissue regeneration in a variety of clinical contexts.
Regarding the region of analysis, we focused on the caudal region of the spinal cord. The caudal region was chosen for analysis because it is the site where the most severe degenerative processes occur after SCI. The caudal region of the spinal cord is particularly vulnerable to damage due to the interruption of the descending pathways, which leads to the loss of voluntary motor control and sensation below the level of the injury. Therefore, it is important to evaluate the effect of MSC-EVs in this region to determine their potential therapeutic value for SCI. We agree that a deeper analysis with other methods/markers would be valuable, and we plan to conduct further studies to investigate additional mechanisms and markers involved in the therapeutic effects of MSC-EVs in spinal cord injury. We tried to emphasize in the title that pilot study: “Intrathecal Injection of Autologous Mesenchymal Stem Cell-Derived Extracellular Vesicles in Spinal Cord Injury: A Feasibility Study in Pigs”
We acknowledge that our study is an exploratory analysis, and we will emphasize this point in our discussion section. We believe that our findings contribute a valuable starting point for further investigations into the mechanisms underlying the therapeutic effects of MSC-EVs in spinal cord injury.

Reviewer 3 Report
In this article, the authors showed that EV from MSCs improved motor function of SCI pigs. However, unfortunately, the novelty of this study doesn’t reach to enough quality for the publication in IJMS. As the authors referred to in the discussion section, several studies already reported the therapeutic effect of EVs from MSCs. Although I understand that this is the first study of SCI pigs treated with EV from MSCs. However, again, I don’t think that it is enough for the strong point of the publication of IJMS. Therefore, I cannot recommend this article for the publication.
Thank you for giving me an opportunity to review this article.
Author Response
In this article, the authors showed that EV from MSCs improved motor function of SCI pigs. However, unfortunately, the novelty of this study doesn’t reach to enough quality for the publication in IJMS. As the authors referred to in the discussion section, several studies already reported the therapeutic effect of EVs from MSCs. Although I understand that this is the first study of SCI pigs treated with EV from MSCs. However, again, I don’t think that it is enough for the strong point of the publication of IJMS. Therefore, I cannot recommend this article for the publication.
Thank you for giving me an opportunity to review this article.
Authors: We would like to thank the reviewer for this review and expressing his point of view. We hope that our responses will satisfy the reviewer. Хотелось бы сообщить, что в соответствии с замечаниями других двух рецензентов, мы внесли некоторые изменения в манускрипт. Мы надеемся, что следующие корректировки, возможно, поменяют мнение уважаемого рецензента. Вот список изменений, внесенных в последнюю версию манускрипта в соответствии с замечаниями рецензентов:
- The manuscript has been corrected by a proficient English speaker. We have highlighted all the changes in the manuscript in red.
- Regarding the region of analysis, we focused on the caudal region of the spinal cord. The caudal region was chosen for analysis because it is the site where the most severe degenerative processes occur after SCI. The caudal region of the spinal cord is particularly vulnerable to damage due to the interruption of the descending pathways, which leads to the loss of voluntary motor control and sensation below the level of the injury. Therefore, it is important to evaluate the effect of MSC-EVs in this region to determine their potential therapeutic value for SCI. We agree that a deeper analysis with other methods/markers would be valuable, and we plan to conduct further studies to investigate additional mechanisms and markers involved in the therapeutic effects of MSC-EVs in spinal cord injury. We tried to emphasize in the title that pilot study: “Intrathecal Injection of Autologous Mesenchymal Stem Cell-Derived Extracellular Vesicles in Spinal Cord Injury: A Feasibility Study in Pigs”. We acknowledge that our study is an exploratory analysis, and we will emphasize this point in our discussion section. We believe that our findings contribute a valuable starting point for further investigations into the mechanisms underlying the therapeutic effects of MSC-EVs in spinal cord injury.
- In response to the reviewer's comment, we acknowledge that there may be some key information missing regarding the animal model used in this study. We ensured that this information is provided in the revised manuscript, including details on the age and sex of the pigs used, the method of SCI induction, and the sample size used in each experimental group. Regarding the organization of the results, we carefully reviewed the manuscript and consider any changes that could improve the clarity and flow of the presentation. We appreciated the reviewer's feedback and added their comments into account in order to improve the manuscript. The updated text is red in pdf.
- We have now updated the introduction section to provide a more detailed description of the structure, functions, and current applications of extracellular vesicles (EVs). We hope that the new information better clarifies the importance and potential of EVs in various fields, including diagnostic and therapeutic applications. We have added it in Introduction “Exosomes are the smallest type….”.
- We have added the necessary information about the differences and functions of exosomes and microvesicles in Introduction “The main difference between exosomes and EVs…”. Exosomes are formed by the inward budding of endosomal membranes and play important roles in intercellular communication and regulation of physiological processes, while microvesicles are formed by the outward budding of the plasma membrane and are involved in cellular processes such as cellular signaling, cell-to- cell communication, and the release of cellular waste and debris.
- We added Table 1. In this table, we showed in detail the morphological charactristics eof extracellular vesicles produced by different protocols in the laboratory. If you are asking whether we compared extracellular vesicles obtained with cytolysin B to natural ones, we did not perform such a comparison in our study. However, previous literature has shown that the sizes of laboratory-derived extracellular vesicles and naturally occurring ones are comparable.
- We have added Figure 1 with EVs production scheme. We have also added a table with characterization of obtained MSCs-derived EVs based on electron microscopy analysis as the report of the EVs from the three procedures according to the defined selection criteria.
- We added the full information about the injury model and EV administration in Materials and Methods.

Round 2
Reviewer 2 Report
Thanks for the modifications. I think the article has improved a lot.
Author Response
Dear reviewer, thank you for reviewing our work and for your comments and suggestions!
Reviewer 3 Report
Unfortunately, even after the author’s great effort, I cannot find this article is worth for publication simply because of lacking of the novelty. Again, I appreciate for giving me an opportunity to review this article.

Author Response
Authors: Thank you for your review and feedback. We appreciate your time and effort in evaluating our article. While we understand your concerns regarding the novelty of our study, we believe that our findings contribute to the growing body of literature on the use of MSCs-derived EVs for spinal cord injury. We have carefully chosen an optimal method for obtaining EVs and performed a qualitative assessment of the EVs, which we believe adds value to the field. We hope that you will reconsider our article for publication. The novelty of this study is that it evaluates the therapeutic potential of intrathecal injection of autologous MSCs-derived EVs in the subacute period of pig contused SCI for the first time. The study also chose an optimal method for obtaining cytochalasin B-induced EVs and performed a qualitative assessment of the EVs.